# SLC38A5 Modulates Ferroptosis to Overcome Gemcitabine Resistance in Pancreatic Cancer

**DOI:** 10.3390/cells12202509

**Published:** 2023-10-23

**Authors:** Myeong Jin Kim, Hyung Sun Kim, Hyeon Woong Kang, Da Eun Lee, Woosol Chris Hong, Ju Hyun Kim, Minsoo Kim, Jae-Ho Cheong, Hyo Jung Kim, Joon Seong Park

**Affiliations:** 1Department of Surgery, Gangnam Severance Hospital, Yonsei University College of Medicine, Seoul 06229, Republic of Korea; audwls8739@gmail.com (M.J.K.); milky8508@yuhs.ac (H.S.K.); kanghw9305@gmail.com (H.W.K.); hsgtj06321@gmail.com (D.E.L.); chrish95@gmail.com (W.C.H.); juhyun9503@gmail.com (J.H.K.); alstndi777@gmail.com (M.K.); 2Brain Korea 21 PLUS Project for Medical Science, Yonsei University, Seoul 03722, Republic of Korea; jhcheong@yuhs.ac; 3Yonsei University College of Medicine, Seoul 06229, Republic of Korea; 4Department of Surgery, Yonsei University College of Medicine, Seoul 06229, Republic of Korea

**Keywords:** SLC38A5, gemcitabine resistance, PDAC, lipid ROS, ferroptosis

## Abstract

Pancreatic cancer is characterized by a poor prognosis, with its five-year survival rate lower than that of any other cancer type. Gemcitabine, a standard treatment for pancreatic cancer, often has poor outcomes for patients as a result of chemoresistance. Therefore, novel therapeutic targets must be identified to overcome gemcitabine resistance. Here, we found that SLC38A5, a glutamine transporter, is more highly overexpressed in gemcitabine-resistant patients than in gemcitabine-sensitive patients. Furthermore, the deletion of SLC38A5 decreased the proliferation and migration of gemcitabine-resistant PDAC cells. We also found that the inhibition of SLC38A5 triggered the ferroptosis signaling pathway via RNA sequencing. Also, silencing SLC38A5 induced mitochondrial dysfunction and reduced glutamine uptake and glutathione (GSH) levels, and downregulated the expressions of GSH-related genes NRF2 and GPX4. The blockade of glutamine uptake negatively modulated the mTOR-SREBP1-SCD1 signaling pathway. Therefore, suppression of SLC38A5 triggers ferroptosis via two pathways that regulate lipid ROS levels. Similarly, we observed that knockdown of SLC38A5 restored gemcitabine sensitivity by hindering tumor growth and metastasis in the orthotopic mouse model. Altogether, our results demonstrate that SLC38A5 could be a novel target to overcome gemcitabine resistance in PDAC therapy.

## 1. Introduction

Pancreatic cancer is a lethal and aggressive cancer that has a five-year survival rate of less than 10% [1]. Because it is difficult to detect, most patients have pancreatic cancer at an advanced stage, which lowers the success rate of surgical treatment. Therefore, these patients tend to receive chemotherapy [2,3]. Currently, gemcitabine is one of the most used medications for the treatment of patients with pancreatic cancer [4]. However, the effects of gemcitabine are limited because drug resistance occurs within weeks of treatment [5]. Because there are relatively few studies on gemcitabine resistance in pancreatic cancer, a new strategy is necessary to alleviate drug resistance in pancreatic cancer.

Solute carrier family 38 membrane 5 (SLC38A5), also known as sodium-coupled neutral amino acid transporter 5 (SNAT5), transports amino acids such as asparagine, glutamine, methionine, serine, and glycine across the cell membrane [6]. Unlike other amino acid transporters, SLC38A5 mainly transports glutamine into the cancer cells [7,8]. For example, upregulation of SLC38A5 induces cell proliferation by inducing macropinocytosis in triple-negative breast cancer [9]. Moreover, a previous study reported that SLC38A5 initiates pancreatic neuroendocrine tumor formation [10]. However, there have been no studies on the biological mechanisms of SLC38A5 and gemcitabine resistance in pancreatic cancer. According to The Cancer Genome Atlas (TCGA) database, upregulation of SLC38A5 tends to result in a poor survival rate in pancreatic cancer. Therefore, we hypothesized that there could be a crucial role of SLC38A5 in pancreatic cancer.

Glutamine is an important nutrient in cancer metabolism and is involved in the synthesis of proteins and lipids and de novo glutathione production [11]. Cancer cells tend to consume more glutamine than normal cells during cell proliferation and epithelial-to-mesenchymal transition (EMT) [12]. However, the correlation between glutamine and gemcitabine resistance in pancreatic cancer remains unclear. Therefore, we aimed to study the effect of glutamine transported by SLC38A5 in gemcitabine-resistant cancer cells. As glutamine enters, it is converted into glutamate via glutaminolysis in the mitochondria [13]. Eventually, glutamate is secreted from the mitochondria and contributes to glutathione synthesis [14]. Glutathione is an antioxidant agent that regulates reactive oxygen species (ROS) levels in cancer cells [15]. Additionally, glutamine regulates the mTOR-SREBP1 signaling pathway in cancer cells [16]. SCD1 is a downstream target of SREBP1 signaling and plays a role in converting polyunsaturated fatty acids (PUFAs) that induce lipid ROS into monounsaturated fatty acids (MUFAs) [17]. Therefore, glutamine deprivation upregulates lipid ROS levels in pancreatic cancer cells. Hence, we hypothesized that glutamine depletion may increase the effect of gemcitabine on gemcitabine-resistant pancreatic cancer cells.

Ferroptosis is a type of programmed cell death associated with lipid ROS accumulation [18]. Recently, several studies have shown that ferroptosis plays an important role in various cancers, including pancreatic cancer. Previous studies have shown that ferroptosis alleviates sorafenib resistance in hepatocellular carcinoma and cisplatin resistance in head and neck cancers [19,20]. However, the relationship between ferroptosis and gemcitabine resistance in pancreatic cancer remains unclear. Therefore, we examined ROS and lipid ROS-related genes as indicators of ferroptosis [21].

In this study, we observed that glutamine uptake was increased in gemcitabine-resistant pancreatic cancer. Moreover, the upregulated glutamine levels ultimately contribute to the inhibition of ferroptosis, making pancreatic cancer more resistant to gemcitabine. Conversely, gemcitabine resistance was weakened when SLC38A5 expression was suppressed. We also demonstrated that the depletion of SLC38A5 induces mitochondrial dysfunction, increases ROS levels, and decreases mTOR-SREBP1 signaling. These pathways ultimately induced ferroptosis by increasing lipid ROS levels. Moreover, SLC38A5 inhibition effectively sensitized pancreatic cancer cells to gemcitabine treatment, suppressing tumor weight and metastasis in vivo. Taken together, our results suggest that SLC38A5 is a novel therapeutic target for ameliorating gemcitabine resistance in pancreatic cancer.

## 2. Materials and Methods

### 2.1. The Cancer Genome Atlas Analysis

Expression data were obtained from TCGA, and SLC38A5 levels were measured with transcripts per million (TPM) as log2 (TPM + 1) in the Gene Expression Profiling Interactive Analysis (GEPIA) portal. Clinical parameters and survival data were also obtained from the TCGA pancreatic cancer database. Overall survival and disease-specific survival were analyzed using Kaplan–Meier plots. Correlation analyses of indicated genes were evaluated in the GEPIA portal.

### 2.2. Cell Culture

Human pancreatic cancer cell lines (PANC-1 and Capan-1) were purchased from the American Type Culture Collection (ATCC MD, Manassas, VA, USA). PANC-1 and Capan-1 were cultured in DMEM and RPMI 1640, respectively, and supplemented with 10% fetal bovine serum (Biowest, Nuaillé, France) and 1% antibiotic–antimycotic reagent (Gibco, Waltham, MA, USA) at 37 °C and 5% CO_2_. After one passage, gemcitabine was treated with PANC-1 and Capan-1 cells to establish a gemcitabine-resistant cell line. The surviving cells were subcultured and continuously treated with increasing gemcitabine concentrations (0.1 μM to 10 μM) for 3 months. After the gemcitabine-resistant cells stabilized, they were treated with the same concentration of gemcitabine for 3 months.

### 2.3. Patient Tissue mRNA and Protein

Pancreatic tissues were acquired from patients with pancreatic cancer at Gangnam Severance Hospital. mRNA was isolated using the TRIZOL reagent (Sigma-Aldrich, Munich, Germany). Protein samples were extracted using RIPA lysis buffer (Sigma-Aldrich). The study protocol conformed to the ethical guidelines of the Declaration of Helsinki and was approved by the Institutional Review Board (IRB) of Gangnam Severance Hospital (no.3-2021-0414). 

### 2.4. Small-Interfering RNA Transfection 

Small-interfering RNA transfection was conducted as previously described [22]. Briefly, cells were transfected with small-interfering RNA (siRNA) targeting SLC38A5 (Santa Cruz Biotechnology, Dallas, TX, USA) using Lipofectamine^®^ RNAiMAX transfection reagent (Invitrogen, Paisley, UK) in Opti-MEM (Gibco, Waltham, MA, USA) according to the manufacturer’s protocol.

### 2.5. WST Assay

Cells were seeded in 96-well plates at a density of 5 × 10^3^ (PANC-1) and 7 × 10^3^ (Capan-1) cells per well and were incubated for 24 h. Indicated concentrations of gemcitabine were treated and the cells were incubated for 72 h. After 72 h, 10% water-soluble tetrazolium (WST)-1 reagent (DoGenBio, Seoul, Republic of Korea) replaced the growth medium. The absorbance was measured at 450 nm wavelength using the VersaMax microplate reader (Molecular Devices, San Jose, CA, USA). The IC_50_ values were measured using GraphPad Prism version 8.0 (GraphPad Software, San Diego, CA, USA).

### 2.6. Wound Healing Assay

Cells were seeded in 12-well plates at densities of 2 × 10^5^ (PANC-1) and 4 × 10^5^ (Capan-1) per well. The cells were transfected with siRNA the next day, and scratches were made 24 h later. Microscopic images of the migrated cells were captured at the indicated time intervals.

### 2.7. Invasion Assay

An 8 µm pore size Transwell system was coated with Matrigel for 1 h at room temperature. The cells were seeded in 24-well plates at a density of 1 × 10^5^ (PANC-1) and 2 × 10^5^ (Capan-1). siRNA transfection was conducted afterward. The cells then were transferred to the transwell at a density of 0.7 × 10^5^ (PANC-1) and 2.5 × 10^5^ (Capan-1) and incubated for 24 h. The transwell plate was washed with PBS and fixed in 4% paraformaldehyde. The transferred cells were stained with crystal violet (JUNSEI, Tokyo, Japan).

### 2.8. RNA Sequencing

RNA extraction and RNA sequencing were conducted as previously described [23]. TRIzol reagent (Invitrogen) was used to isolate total RNA, and then RNA quality was assessed using Agilent 2100 Bioanalyzer with RNA 6000 Nano Chip (Agilent Technologies, Amstelveen, The Netherlands). RNA was quantified using an ND-2000 Spectrophotometer (Thermo Inc., Waltham, MA, USA). Library construction for control and test RNAs was performed using QuantSeq 3′ mRNA-Seq Library Prep Kit (Lexogen Inc., Wien, Austria). High-throughput sequencing was performed as 75 bp single-end sequencing using NextSeq 500 (Illumina Inc., San Diego, CA, USA). QuantSeq 3 mRNA-Seq reads were aligned using the Bowtie2.

### 2.9. Glutamine Uptake Assay

Cells were seeded in a 6-well plate at a density of 2.5 × 10^5^ (PANC-1) and 5 × 10^5^ (Capan-1) per well. The next day, they were transfected with siRNA and harvested using lysis buffer and prepared for the measurement of glutamine using the Glutamine Detection Assay Kit (ab197011) from Abcam (Cambridge, UK), according to the manufacturer’s instructions. Relative glutamine uptake was normalized to the number of cells in each experimental group.

### 2.10. OCR and ECAR Measurement

Oxygen consumption rate (OCR) and extracellular acidification rate (ECAR) were conducted as previously described [23]. Briefly, cells were seeded onto XF-24 plates at a density of 5 × 10^5^ cells/well for 24 h and treated with siRNA. Then, they were incubated in XF assay media for 1 h at 37 °C in a non-CO_2_ incubator and stressed with sequential addition of 1 µM oligomycin, 2 µM carbonyl cyanide p-(trifluoromethoxy) phenylhydrazone, and a 0.5 µM cocktail of rotenone/antimycin A. The OCR was normalized to the total cellular protein concentration.

### 2.11. Glutathione Level Assay

Cells were seeded in a 6-well plate at a density of 2.5 × 10^5^ (PANC-1) and 5 × 10^5^ (Capan-1) per well. The next day, they were transfected with siRNA and then collected. Glutathione level was measured using the Glutathione Assay Kit (No. 703002) (Cayman Chemical, MI, USA), according to the manufacturer’s protocol. Glutathione concentrations were calculated using a standard curve and normalized to the total protein levels in each sample.

### 2.12. ROS and Lipid ROS Measurement

Cells were plated in a 6-well plate at a density of 2.5 × 10^5^ (PANC-1) and 5 × 10^5^ (Capan-1) per well. The next day, they were transfected with siRNA and treated with Ferrostatin-1 (20 μM) for 48 h. The cells were then incubated in 20 μM 2′7′-dichlorodihydrofluorescein diacetate (DCF-DA; Sigma-Aldrich) and 10uM BODIPY-C11 (D3861, Invitrogen) for 30 min at 37 °C in the dark. They were then washed twice with PBS. The ROS and lipid ROS levels were measured using a FACScanto II flow cytometer (BD Biosciences, Franklin Lakes, NJ, USA). A minimum of 10,000 events were recorded for each sample.

### 2.13. RT-PCR and qPCR

Cells were seeded in a 6-well-plate and incubated for 24 h, and then transfected with siRNA. After isolating the total RNA, the samples were analyzed via RT-PCR using a Maxime RT-PCR premix kit (Intron, Gyeonggi-do, Republic of Korea). RT-qPCR was performed using a SYBR qPCR reaction mix (Applied Biosystems, Foster City, CA, USA). The primer sequences used in this study are listed in Table 1. Relative mRNA expression level was calculated using the 2^−ΔΔCT^ method, with GAPDH as the reference gene. 

### 2.14. Western Blotting

Western blotting was performed as previously described [22]. Cell lysates were separated using sodium dodecyl sulfate–polyacrylamide gel electrophoresis and transferred to polyvinylidene fluoride membranes. After blocking with 5% skim milk for 1 h, the membranes were incubated with primary antibodies (1:1000) at 4 °C overnight, followed by incubation with horseradish peroxidase (HRP)-conjugated secondary antibodies (1:5000) for 1 h. The protein bands were then exposed to an enhanced chemiluminescent HRP substrate (Waltham, MA, USA) and detected using X-ray film. The primary antibodies: GAPDH (# sc-365062), NRF2 (# sc-365949), GPX4 (# sc-166570), SREBP1 (# sc-17755), and SCD1 (# sc-515844) were purchased from Santa Cruz Biotechnology (Dallas, TX, USA). mTOR (# 2972S), Phospho-mTOR (# 5536S), PI3K (# 4292S), pAKT (# 9271S), AKT (# 9272S), pSTAT3 (# 9145S), and STAT3 (# 9139S) were purchased from Cell Signaling Technology (Danvers, MA, USA). SLC38A5 (# ab72717) and RRM1 (# ab226391) were purchased from Abcam. N-cadherin (# 610920) and E-cadherin (# 610181) antibodies were purchased from BD Biosciences (Franklin Lakes, NJ, USA). Horseradish peroxidase-conjugated goat anti-mouse (# 7076S) and HRP-conjugated goat anti-rabbit (#7074S) secondary antibodies were purchased from Cell Signaling Technology. 

### 2.15. Animal Studies

Six-week-old male nude BALB/c mice were purchased from Orient Bio (Tokyo, Japan). To establish the orthotopic mouse model, 3 × 10^6^ PANC-1 cells were injected into the pancreas of each mouse. After two months, the mice were randomized into two sensitive groups (n = 6) and three resistance groups (n = 6). Then, 4.5 × 10^6^ shRNA lentiviral particles (TL506284V, ORIGENE) were injected into the vein. The following week, gemcitabine (10 mg/kg) and ferrostatin-1 (1 mg/kg) were intraperitoneally injected three times a week for 14 days. On the 14th day, the mice were euthanized. The tumors were harvested, weighed, and fixed in 4% paraformaldehyde. Metastases were photographed. Tumor volume was measured using calipers and calculated using the following formula = 0.5 × length × width^2^. 

All animal experimental procedures followed the National Institutes of Health Guide for the Care and Use of Laboratory Animals and were performed with protocols approved by the Institutional Animal Care and Use Committee of the Seoul Yonsei Pharmaceutical University Experimental Animal Center (approval #2019-0104).

### 2.16. Statistical Analysis

Statistical analyses were performed using one-way or two-way analysis of variance (ANOVA) in GraphPad Prism version 8.0. Data are presented as mean ± standard deviation. Statistical significance was indicated (* *p* < 0.05; ** *p* < 0.01; *** *p* < 0.001; # *p* < 0.05; ## *p* < 0.01; ### *p* < 0.001).

## 3. Results

### 3.1. SLC38A5 Correlates with Gemcitabine-Resistant Pancreatic Cancer Patients

To investigate SLC38A5 expression in pancreatic cancer cells, we analyzed mRNA and protein levels in human pancreatic cancer cell lines. We confirmed the expression of SLC38A5 in five PDAC cell lines (Aspc-1, Bxpc-3, Capan-1, PANC-1, and MIA PaCa-2) using RT-PCR and Western blot analysis (Figure 1A). We examined the expression of SLC38A5 in pancreatic cancer using RT-PCR, Western blotting, and immunohistochemical staining. Based on these experiments, we found that pancreatic tumor tissues have higher expression levels of SLC38A5 than normal tissues (Figure 1B,C). Moreover, increased levels of SLC38A5 tended to decrease the overall survival (OS) and disease-specific survival (DSS) of patients with pancreatic cancer (Figure 1D). To verify whether SLC38A5 and gemcitabine resistance are significantly correlated, we confirmed the mRNA expression levels of SLC38A5 as well RRM1, a widely recognized gemcitabine-resistant marker, in both gemcitabine-sensitive and gemcitabine-resistant patient samples [24,25]. Both SLC38A5 and RRM1 mRNA levels were higher in gemcitabine-resistant pancreatic cancer tissues than in gemcitabine-sensitive patient tissues (Figure 1E). Similarly, the protein levels of both SLC38A5 and RRM1 were increased in drug-resistant pancreatic cancer patients compared with those in the drug-sensitive group (Figure 1F). We also constructed a stage plot related to SLC38A5 expression in human data. SLC38A5 expression gradually increased up to stage three; however, no increase was observed in stage four (Appendix A). We hypothesized that SLC38A5 is overexpressed in gemcitabine-resistant conditions and plays an important role in the regulation of gemcitabine-resistant environments.

### 3.2. Deletion of SLC38A5 Downregulates Cell Viability and Migration in Gemcitabine-Resistant Pancreatic Cancer Cells

To elucidate the role of SLC38A5 in gemcitabine resistance, we established gemcitabine-resistant pancreatic cancer cell lines using PANC-1 and Capan-1 cells (PANC-GR and Capan-GR, respectively) by confirming the half-maximal inhibitory concentration (IC50) via a cell viability test (Figure 2A and Appendix AA). However, the expression of SLC38A5 was not increased in the other GR cell lines (Aspc-1, Bxpc-3, and MIA PaCa-2). Consistent with previous results, higher SLC38A5 and RRM1 expression was confirmed at the mRNA and protein levels in the gemcitabine-resistant cell lines than in the gemcitabine-sensitive cell lines (Figure 2B). To investigate whether SLC38A5 modulates the sensitivity of GR cell lines to gemcitabine, a WST assay was performed in the presence of gemcitabine (10 μM). Gemcitabine-resistant cells were more proliferative than gemcitabine-sensitive cells. Conversely, when SLC38A5 was deleted in both sensitive and resistant cells, each cell type showed decreased proliferation according to both cell viability and RT-PCR (Figure 2C). Previous studies have shown that gemcitabine-resistant pancreatic cancer cells show increased levels of both migration and invasion compared with gemcitabine-sensitive cells [26,27]. However, it remains unclear whether SLC38A5 plays a significant role in cancer metastasis. Knockdown of SLC38A5 expression hindered both the migration and invasion abilities of cancer cells in the wound healing and transwell invasion assays (Figure 2D,E). To verify whether SLC38A5 controls epithelial-to-mesenchymal transition (EMT), we confirmed EMT-related genes by regulating SLC38A5 mRNA and protein expression in GR cell lines. E-cadherin, an epithelial marker, increased in SLC38A5 knockdown cells. Conversely, the expression of N-cadherin, a mesenchymal marker, decreased (Figure 2F,G). We also analyzed the gene set enrichment analysis (GSEA) results using RNA sequencing data. SLC38A5 deletion was found to reduce wound healing ability, EMT, and angiogenesis in PANC-GR cells (Appendix AD). Therefore, we confirmed that SLC38A5 is involved in the proliferation and EMT of gemcitabine-resistant pancreatic cancer cells.

### 3.3. Overview of Enrichment Analyses between Gemcitabine-Resistant Cells and SLC38A5 Inhibition

Next, we performed RNA sequencing to determine whether SLC38A5 plays a crucial role in any biological mechanism in gemcitabine-resistant cells. Using heat map data, we identified various differential genes compared among PANC-1, PANC-GR, and PANC-GR cells with SLC38A5 knockdown (Figure 3A). Using the DAVID analysis tool, a bubble plot showed that inhibition of SLC38A5 upregulated ferroptosis and lipid oxidation (Figure 3B). Furthermore, the TCA cycle, hypoxia, and cell cycle pathways are downregulated [28,29]. Additionally, we analyzed the inhibition of SLC38A5 in promoting ferroptosis through the GSEA graph for both PANC-GR and Capan-GR (Figure 3C). Ferroptosis is a programmed cell death pathway associated with lipid ROS. We performed a correlation analysis of the expression of SLC38A5 and ferroptosis-related genes. We found that SLC7A11 and FTH, ferroptosis-inhibiting genes, were positively correlated with SLC38A5, whereas ACSL4 and ALOX12, ferroptosis-inducing genes, were negatively correlated with SLC38A5 [30,31] (Figure 3D). Also, we examined similar results via q-PCR analysis compared among PANC-1, PANC-GR, and PANC-GR with SLC38A5 knockdown (Figure 3E). Collectively, we found that SLC38A5 inhibition in PANC-GRs induced lipid ROS and ferroptosis. We believe that the results of this study will play an important role in overcoming gemcitabine resistance.

### 3.4. SLC38A5 Modulates Lipid ROS through GSH-Mediated ROS and mTOR-SREBP1 Signaling in Gemcitabine-Resistant Pancreatic Cancer Cells

Cancer cells, specifically drug-resistant cells, rely more on nutrients than normal cells because of their rapid cell proliferation and migration. Glutamine is the main fuel used by cancer cells. Therefore, we examined whether SLC38A5 regulates glutamine consumption in gemcitabine-resistant cells using a glutamine assay kit. We found that the glutamine consumption rate was much higher in gemcitabine-resistant cells (PANC-GR and Capan-GR) than in gemcitabine-sensitive cells (PANC-1 and Capan-1). Additionally, its rate was downregulated when SLC38A5 was silenced (Figure 4A and Appendix AA). When glutamine enters cells, it is involved in mitochondrial functions, such as the TCA cycle, is converted to glutamate through glutaminolysis, and is then released from the mitochondria [13]. Therefore, we measured the oxygen consumption rate (OCR) and extracellular acidification rate (ECAR) using a Seahorse XF24 extracellular flux analyzer to confirm mitochondrial activity. Both OCR and ECAR levels were lower in PANC-GR cells with SLC38A5 deletion (Figure 4B). Additionally, GSEA showed that the inhibition of SLC38A5 reduced mitochondrial activity (Figure 4C and Appendix AB). The presence of glutamate in cells contributes to GSH synthesis [32]. Glutathione is a key factor in the ROS levels related to ferroptosis [33]. Using a glutathione assay kit, glutathione levels were found to be higher in gemcitabine-resistant cells than in gemcitabine-sensitive cells. Conversely, when SLC38A5 was suppressed, glutathione levels decreased (Figure 4D and Appendix AC). We further confirmed that the GSH-related genes NRF2 and GPX4 were upregulated in PANC-GR and downregulated in PANC-GR transfected with siSLC38A5 via RT-PCR and Western blotting (Figure 4E,F). To investigate whether SLC38A5 regulates intercellular ROS levels, we performed fluorescence-activated cell sorting (FACS) analysis using DCF-DA staining by manipulating the expression levels of SLC38A5 in GR cell lines. We observed that ROS levels were much lower in gemcitabine-resistant cell lines than in gemcitabine-sensitive cell lines. As SLC38A5 expression decreased, ROS levels also increased (Figure 4G and Appendix AE). Based on the previous results from FACS analysis, we also confirmed via GSEA that ROS levels increased when SLC38A5 expression levels were suppressed (Figure 4H and Appendix AF). Previous studies have reported that glutamine uptake leads to upregulation of the mTOR-SREBP1 signaling pathway, which regulates lipid ROS [16]. SREBP1 (sterol regulatory element-binding protein 1) is one of the proteins regulating lipid metabolism and acts as a transcription factor for lipid synthesis-related genes such as FASN, ACLY, and SCD1 [34]. Recent studies have demonstrated that SCD1 accumulates monounsaturated fatty acids (MUFAs), which suppress and inhibit ferroptosis [17,35]. Both the mRNA and protein levels of SREBP1 and SCD1 were increased in gemcitabine-resistant cells and were downregulated when SLC38A5 was silenced (Figure 4I,J and Appendix AG). Prior to this experiment, we confirmed that mTOR phosphorylation was reduced in a glutamine-deficient environment by culturing cells in glutamine-free media. Western blotting showed that phospho-mTOR levels decreased in glutamine-free media (Appendix AI). To determine whether SLC38A5 significantly regulates ferroptosis by controlling lipid ROS levels, we measured lipid ROS levels in gemcitabine-resistant cell lines by manipulating SLC38A5 expression via FACS analysis with staining BODIPY-C11. As a result, lipid ROS levels were decreased in gemcitabine-resistant cells but were stored when SLC38A5 expression levels were inhibited (Figure 4K and Appendix AH). Gemcitabine induces apoptosis; however, this apoptotic effect is weakened under gemcitabine-resistant conditions [36,37]. To investigate whether SLC38A5 regulates apoptosis, we measured the mRNA expression levels of apoptosis-related genes in PANC-GR cells. PANC-GR showed more anti-apoptotic effects than PANC-GS based on the downregulated levels of BAX, caspase 3, and caspase 9 than PANC-GS. Conversely, the mRNA levels of the apoptosis-inducing genes were upregulated when SLC38A5 was silenced (Figure 4L). Thus, glutamine uptake through SLC38A5 ultimately suppresses lipid ROS production through the GSH-mediated ROS and mTOR-SREBP1 signaling pathway in gemcitabine-resistant cells, leading to apoptosis.

### 3.5. Inhibition of SLC38A5 Induces Ferroptosis in Gemcitabine-Resistant Pancreatic Cancer Cells

To validate whether SLC38A5 directly regulates ferroptosis, we used ferrostatin-1 as a ferroptosis inhibitor. Inhibition of SLC38A5 sensitized PANC-GR cells to gemcitabine (Figure 2C); however, their viability was rescued when ferrostatin-1 was applied to PANC-GR cells, even in the presence of Erastin, a ferroptosis inducer (Figure 5A). With respect to cell proliferation, we also confirmed that cell proliferation-related genes were upregulated when ferroptosis was inhibited (Figure 5B). Using wound healing and invasion assays, we also confirmed that ferrostatin-1 attenuated the EMT effect of siSLC38A5 in PANC-GR cells (Figure 5C,D). We also utilized NRF2 and GPX4 as antioxidant markers to verify whether ferrostatin-1 could regulate when GR cells express a low level of SLC38A5 at both mRNA and protein levels. We confirmed that ferrostatin-1 increased the antioxidant levels (Figure 5E,F). Moreover, ferrostatin-1 inhibits ferroptosis by inhibiting lipid ROS [38,39]. Consistent with previous studies, our results showed that ferrostatin-1 inhibited both ROS and lipid ROS production (Figure 5G,H). Therefore, the anti-proliferative, EMT, ROS, and ferroptotic effects mediated by SLC38A5 inhibition could be prevented when ferroptosis is suppressed.

### 3.6. Suppression of SLC38A5 Inhibits Tumor Growth and Metastasis in Orthotopic Model

To evaluate whether SLC38A5 plays an important role in the metastasis of gemcitabine-resistant pancreatic tumors in vivo, we generated a BALB/c nude mouse model bearing orthotopic tumors. After 2 months, gemcitabine and ferrostatin-1 were intraperitoneally injected at the indicated concentrations three times a week for 2 weeks (Figure 6A). After drug injection, each group exhibited significantly different tumor sizes. The gemcitabine-resistant group had a much larger tumor size than the gemcitabine-sensitive group. Moreover, inhibition of SLC38A5 hindered tumor growth in both the gemcitabine-sensitive and gemcitabine-resistant groups. Furthermore, treatment with ferrostatin-1 increased tumor size and weight (Figure 6B). Expression levels of SLC38A5, EMT markers, and ferroptosis markers were reduced in the tumor and livers of the knockdown group in accordance with the in vitro results; the treatment of Ferrostatin-1, however, negated the effect of the knockdown (Figure 6C). Western blot revealed the same trend in protein expressions in the tumors as well (Figure 6D). Furthermore, the gemcitabine-resistant group showed liver metastases, whereas the gemcitabine-resistant with SLC38A5 knockdown group showed no liver metastases. Conversely, the inhibition of ferroptosis-induced liver metastasis prevented SLC38A5 inhibition (Figure 6E). 

Collectively, SLC38A5 inhibition sensitizes drug-resistant pancreatic cancer cells to gemcitabine by inducing ferroptosis and suppressing tumor growth and metastasis. Therefore, targeting SLC38A5 may play a major role in improving the 5-year survival rate of patients with pancreatic cancer and overcoming chemoresistance in pancreatic cancer for clinical purposes. 

## 4. Discussion

Pancreatic cancer is one of the most fatal diseases with one of the highest mortality rates among all major cancers [40]. Despite the development of many treatments and diagnostic methods, the poor prognosis of pancreatic cancer has not improved compared with that of other cancers. Although gemcitabine is known as a standard drug for the treatment of PDAC, it has a major limitation in that drug resistance can easily develop. Therefore, the present study aimed to overcome gemcitabine resistance in pancreatic cancer.

SLC38A5 is a well-known amino acid transporter that transports glutamine [8]. Recent studies have shown that glutamine plays a critical role in the growth, survival, and drug resistance of pancreatic cancer [41]. Glutamine is a key nutrient that promotes drug resistance in breast cancer [42]. However, the mechanism by which glutamine affects drug resistance in pancreatic cancer has not yet been elucidated. In this study, we investigated the mechanism by which glutamine regulates gemcitabine resistance. A previous study has shown that glutamine is transported into the mitochondria and converted into glutamate via glutaminolysis. It is also involved in GSH synthesis [11]. We confirmed that glutamine uptake and glutathione levels were increased in gemcitabine-resistant PDAC cells and decreased when SLC38A5 was silenced (Figure 4A,D), suggesting that glutamine transported by SLC38A5 contributes to glutathione synthesis in PDAC cells. Synthesized glutathione regulates intracellular ROS levels by modulating the related genes NRF2 and GPX4. Many studies indicate that reduced ROS levels may also induce gemcitabine resistance and are characteristic of gemcitabine resistance [43,44]. We also confirmed reduced ROS levels in gemcitabine-resistant PDAC cells (PANC-GR and Capan-GR) using FACS analysis. Conversely, the ROS levels increased when SLC38A5 was inhibited (Figure 4G and Appendix AE).

Several studies have demonstrated that glutamine increases mTOR activity through phosphorylation in cancer cells [16]. We investigated that phosphoryl-mTOR was downregulated in the glutamine-free medium of gemcitabine-resistant PDAC cells (Appendix AI). We found that phosphoryl-mTOR was upregulated in gemcitabine-resistant PDAC cells compared with that in sensitive cells and was downregulated when SLC38A5 was inhibited (Figure 4J). SREBP1 is known to be involved in fatty acid synthesis and is a downstream target of mTOR [45]. SREBP1 and SCD1 are inhibitory markers of ferroptosis because they convert polyunsaturated fatty acids (PUFAs), which trigger lipid oxidation, to MUFAs [35]. We showed that the mTOR-SREBP1-SCD1 pathway was upregulated in gemcitabine-resistant PDAC cells but decreased when SLC38A5 was silenced (Figure 4I,J). Lipid ROS production is known to be mediated by several pathways [46]. We elucidated the lipid ROS levels through two pathways: the GSH-mediated ROS level and the mTOR pathway. We validated that lipid ROS are downregulated in gemcitabine-resistant PDAC cells compared with gemcitabine-sensitive cells. Furthermore, lipid ROS levels increased when SLC38A5 was silenced (Figure 4K and Appendix AH).

Ferroptosis is a type of cell death triggered by the accumulation of lipid ROS [18]. Our results indicate that SLC38A5 deletion results in the accumulation of lipid ROS in gemcitabine-resistant PDAC cells. In the present study, we used ferrostatin-1, which inhibits ferroptosis by preventing lipid ROS production. As hypothesized, ferrostatin-1 treatment had the opposite effect on SLC38A5 inhibition in gemcitabine-resistant PDAC cells (Figure 5). Based on our previous results, we showed that suppression of ferroptosis in gemcitabine-resistant PDAC cells was triggered by SLC38A5 inhibition.

In conclusion, the upregulation of SLC38A5 in gemcitabine-resistant PDAC cells increased the viability and invasive ability of gemcitabine-resistant PDAC cells by upregulating glutamine uptake. SLC38A5 significantly reduced ferroptosis through the inhibition of lipid ROS accumulation in gemcitabine-resistant PDAC cells. However, SLC38A5 silencing in gemcitabine-resistant PDAC cells reduced cell proliferation and EMT and induced ferroptosis. These mechanistic changes eventually increased the sensitivity of cells resistant to gemcitabine. In summary, we believe that targeting SLC38A5 will significantly contribute to the treatment of patients with gemcitabine-resistant pancreatic cancer and improve their 5-year survival rate.

## 5. Conclusions

Overall, our study showed that inhibition of SLC38A5 reduced gemcitabine resistance in PDAC. Glutamine uptake due to SLC38A5 inhibition lowered GSH levels and increased intracellular ROS levels. Also, deletion of SLC38A5 downregulated mTOR-SREBP1-SCD1 signaling. Inhibition of SLC38A5 increased lipid ROS level and led to ferroptosis, suppressing tumor growth and metastasis in vivo. Therefore, targeting SLC38A5 may be a promising therapeutic agent for improving the survival rate of patients with gemcitabine-resistant pancreatic cancer.

## Figures and Tables

**Figure 1 cells-12-02509-f001:**
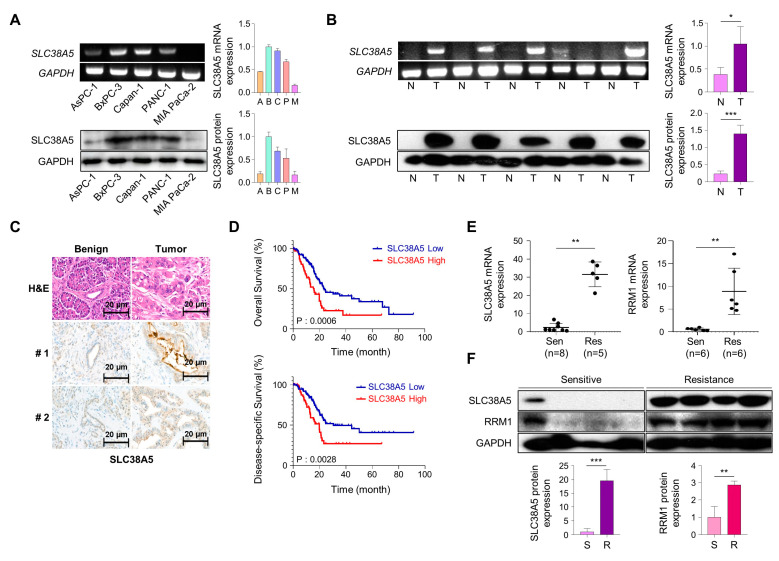
SLC38A5 correlates with gemcitabine-resistant pancreatic cancer patients. (**A**) mRNA and protein expression of SLC38A5 in pancreatic cancer cell lines. (**B**) mRNA and protein expression of SLC38A5 in normal tissues and pancreatic cancer tissues. (**C**) H&E staining of the SLC38A5 in tumor tissue. (**D**) Kaplan–Meier analysis for overall survival, and disease-specific survival between the SLC38A5 high group and the low group. (**E**) Dot plot of SLC38A5 and RRM1 gene expression level in purified mRNA from gemcitabine-sensitive and gemcitabine-resistant pancreatic cancer patient tissue. (**F**) Protein expression of SLC38A5 and RRM1 between gemcitabine-sensitive and gemcitabine-resistant pancreatic cancer patient tissue. All experiments were performed at least three times. Bars represent means ± SD. *, *p* < 0.05; **, *p* < 0.01; ***, *p* < 0.001.

**Figure 2 cells-12-02509-f002:**
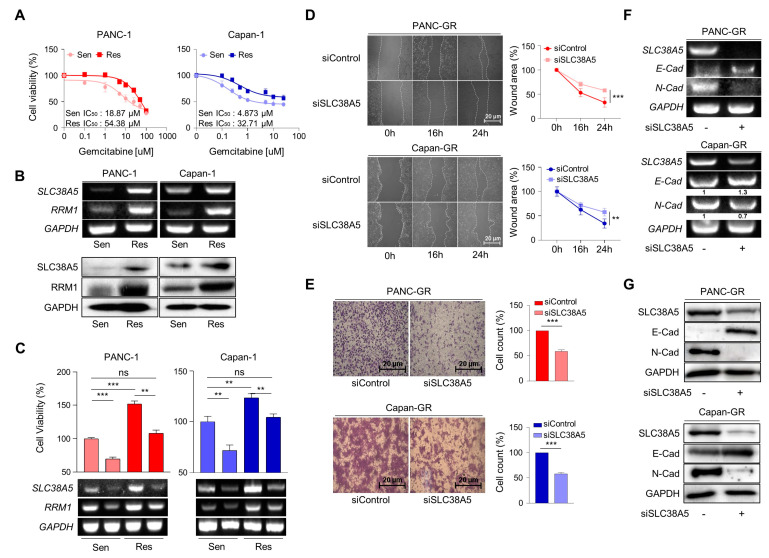
Deletion of SLC38A5 downregulates cell viability and migration in gemcitabine-resistant pancreatic cancer cells. (**A**) Median inhibitory concentration (IC50) values of gemcitabine for each cell line (PANC-1 and Capan-1). (**B**) mRNA and protein expression of SLC38A5 and RRM1 in gemcitabine-sensitive PDAC cell lines (PANC-1 and Capan-1) and gemcitabine-resistant PDAC cell lines (PANC-GR and Capan-GR). (**C**) Cell viability of PANC-1, PANC-GR, Capan-1, and Capan-GR was determined using the WST assay. Each cell line was transfected with siSLC38A5. Transfection was confirmed via RT-PCR. (**D**) Migration of cells was measured via wound healing assay. The statistical analysis of wound closure percentage was measured using Image J software (version 1.8.0). (**E**) Representative photographs of the number of invaded cells that pass through the Matrigel-coated Transwell invasion assay. (**F**,**G**) The EMT marker expression of siRNA transfected PANC-GR and Capan-GR was analyzed via RT-PCR (**F**) and Western blot (**G**). All experiments were performed at least three times. Bars represent means ± SD. ns = not significant; **, *p* < 0.01; ***, *p* < 0.001.

**Figure 3 cells-12-02509-f003:**
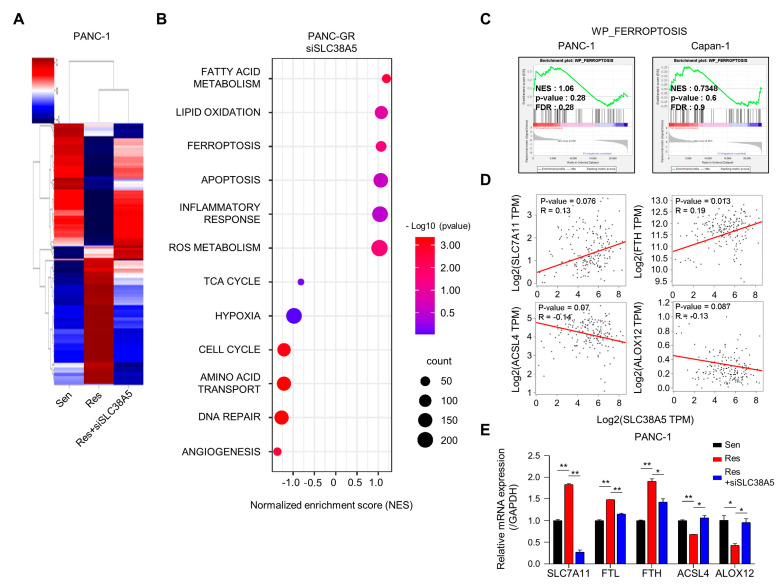
Inhibition of SLC38A5 induces changes in ferroptosis-regulating genes. (**A**) Heat map analysis of the distribution of genes from PANC-1, PANC-GR, and PANC-GR with siSLC38A5 cells. (**B**) DAVID-based gene ontology bubble plot analysis of mRNA sequencing results from PANC-GR and PANC-GR with siSLC38A5 cells. (**C**) GSEA for significant enrichment for ferroptosis. (**D**) Correlation analysis of SLC38A5 and ferroptosis-related genes in pancreatic cancer. (**E**) mRNA levels of the ferroptosis-related genes were quantified via qPCR in PANC-1, PANC-GR, and PANC-GR with siSLC38A5. All experiments were performed at least three times. Bars represent means ± SD. *, *p* < 0.05; **, *p* < 0.01.

**Figure 4 cells-12-02509-f004:**
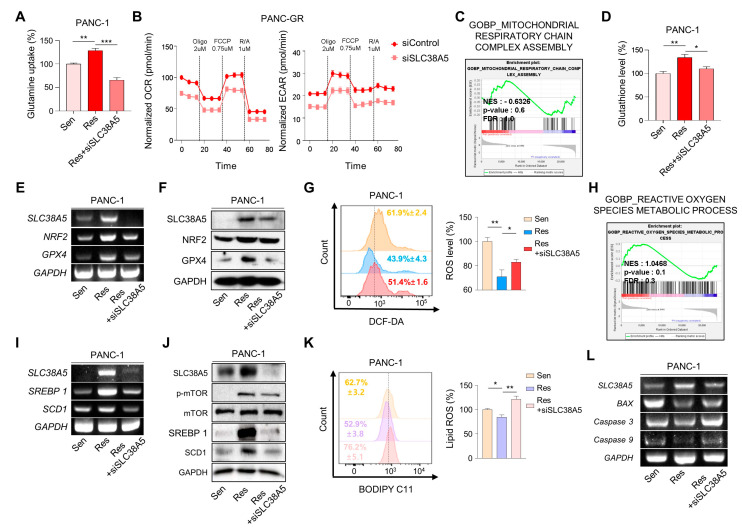
SLC38A5 modulates lipid ROS through GSH-mediated ROS and mTOR-SREBP1 signaling in gemcitabine-resistant pancreatic cancer cells. (**A**) Comparison of glutamine uptake of PANC-1, PANC-GR, and PANC-GR with siSLC38A5. (**B**) OCR and ECAR of PANC-GR cells (siControl and siSLC38A5) were determined and calculated. (**C**) GSEA for significant enrichment for mitochondrial activity. (**D**) Comparison of glutathione levels of PANC-1, PANC-GR, and PANC-GR with siSLC38A5. (**E**,**F**) Expression of glutathione-related genes in PANC-1, PANC-GR, and PANC-GR with siSLC38A5 was analyzed via RT-PCR (**E**) and Western blot (**F**). (**G**) Relative ROS production in PANC-1, PANC-GR, and PANC-GR with siSLC38A5 was measured using flow cytometry after dichlorofluorescein (DCF) staining. The bar graph was analyzed using Image J software. (**H**) GSEA for significant enrichment for ROS level. (**I**,**J**) Expression of mTOR-SREBP1 signaling genes was analyzed via RT-PCR (**I**) and Western blot (**J**). (**K**) Flow cytometry analysis of lipid ROS in PANC-1, PANC-GR, and PANC-GR with siSLC38A5. The bar graph was analyzed using Image J software. (**L**) mRNA levels of apoptosis-related genes in PANC-1, PANC-GR, and PANC-GR with siSLC38A5. All experiments were performed at least three times. Bars represent means ± SD. *, *p* < 0.05; **, *p* < 0.01; ***, *p* < 0.001.

**Figure 5 cells-12-02509-f005:**
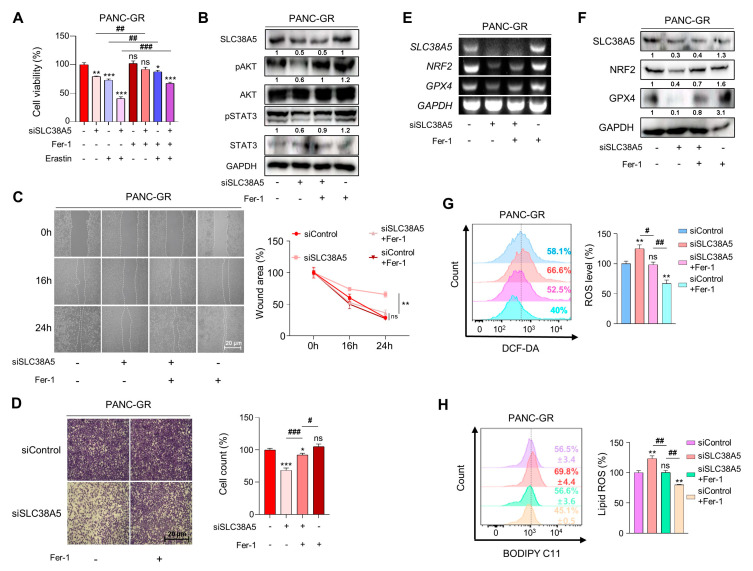
Inhibition of SLC38A5 induces ferroptosis in gemcitabine-resistant pancreatic cancer cells. (**A**) A WST assay was performed to determine cell viability using indicated PANC-GR groups. (**B**) The cell proliferation marker was confirmed via Western blot. (**C**,**D**) Migration of cells was measured through wound healing assay (**C**) and Matrigel-coated transwell invasion assay (**D**). (**E**,**F**) Expression of glutathione-related genes was measured via RT-PCR (**E**) and Western blot (**F**). (**G**) Relative ROS production in indicated PANC-GR groups was measured via FACS analysis. (**H**) FACS analysis of lipid ROS in indicated PANC-GR groups. The bar graph was analyzed using Image J software. An amount of 20 μM of Ferrostatin-1 and 10 μM of Erastin were treated in the control and knockdown groups. All experiments were performed at least three times. Bars represent means ± SD. ns = not significant; *, *p* < 0.05; **, *p* < 0.01; ***, *p* < 0.001, #, *p* < 0.05, ##, *p* < 0.01, ###, *p* < 0.001.

**Figure 6 cells-12-02509-f006:**
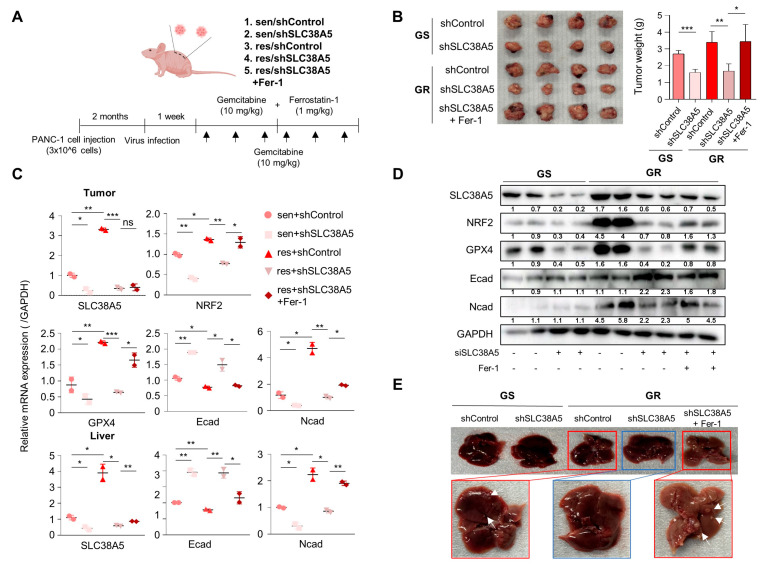
Knockdown of SLC38A5 suppresses tumor growth and metastasis in vivo. (**A**) Schematic of mouse experimental grouping and processing of the orthotopic model. (**B**) Images and weights of the tumors harvested from mice. (**C**) mRNA levels of the EMT and ferroptosis-related genes in tumors and livers were quantified via qPCR. (**D**) Protein expression of EMT and ferroptosis-related genes in tumors. (**E**) Liver metastasis images. The white arrows: liver metastasis. All experiments were performed at least three times. Bars represent means ± SD. ns = not significant; *, *p* < 0.05; **, *p* < 0.01; ***, *p* < 0.001.

**Table 1 cells-12-02509-t001:** List of primers.

Name	Primer Sequence (5′→3′)
GAPDH F GAPDH R SLC38A5 F SLC38A5 R RRM1 F RRM1 R E-Cadherin F E-Cadhern R N-Cadhenn F N-Cadhenin R SLC7A11 F SLC7A11 R FTL F FTL R FTH F FTH R ACSL4 F ACSL4 R ALOX12 F ALOX12 R NRF2 F NRF2 R GPX4 F GPX4 R SREBP1 F SREBP1 R SCD1 F SCD1 R BAX F BAX R Caspase 3 F Caspase 3 R Caspase 9 F Caspase 9 R Mouse GAPDH F Mouse GAPDH R Mouse SLC38A5 F Mouse SLC38A5 R Mouse NRF2 F Mouse NRF2 R Mouse GPX4 F Mouse GPX4 R Mouse E-Cadhenn F Mouse E-Cadhenn R Mouse N-Cadherin F Mouse N-Cadherin R	GTCTCCTCTGACTTCAACAGCG ACCACCCTGTTGCTGTAGCCAA GTTGGGGCCATGTCCAGTTA AGTGTTTCATGAGGGCGAGG TCTCAGACGGAAACAGGCAC GCACAGGTTGCTGCATTTGA GCTCCTGAAAAGAGAGTGGAAG TGGCAGTGTCTCTCCAAATCCG CCTCCAGAGTTTACTGCCATGAC GTAGGATCTCCCCCACTGATTC TGGTCAGAAAGCCTGTTGTGT TGCTCCAATGATGGTGCCAA AAAGCTGAACCAGGCCCTTT CGAAGAGTACTCGCCCAGC AGCTCTACGCCTCCTACGTT AAGGAAGATTCGGCCACCTC TGGTTCTACTGGCCGACCTA TAGCACATGAGCCAAAGGCA GTCAACACAGGCCAGATGGA TGATGCACGTGGTCTTCACA TCCAGTCAGAAACCAGTGGAT GAATGTCTGCGCCAAAAGCTG AGAGATCAAAGAGTTCGCCGC TCTTCATCCACTTCCACAGCG TTCTCACCTCCCAGCTCTGT AGGTGAGACGTGCCAGACTT TTCGTTGCCACTTTCTTGCG CCGGGGGCTAATGTTCTTGT TCAGGATGCGTCCACCAAGAAG TGTGTCCACGGCGGCGGCAATCATC GGAAGCGAATCAATGGACTCTGG GCATCGACATCTGTACCAGACC GTTTGAGGACCTTCGACCACCT CAACGTACCAGGAGGCACTCTT TGTGAACGGATTTGGCCGTA ACTGTGCCGTTGAATTTGCC TGGCACACACTGGAGTCATC ACGGATGCCTACAACACTGG AACAGAACGGCCCTAAAGCA TGGGATTCACGCATAGGAGC CTAGTCGATCTGCATGCCCG GGCATCGTCCCCATTTACAC CAACGATCCTGACCAGCAGT TGTATTGCTGCTTGGCCTCA AGCCAACCTAACTGTCACGG ATTGCTGTTGGGGTCTGTCA

## Data Availability

The datasets used and analyzed in this paper are available from the corresponding author upon reasonable request.

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
