# Peer review of "SLC38A5 Modulates Ferroptosis to Overcome Gemcitabine Resistance in Pancreatic Cancer"

_cells, 2023, doi:10.3390/cells12202509_

Round 1

Reviewer 1 Report

Major comments

1.     The data in Figure 3 collectively shows that genes related to ferroptosis regulation are differentially expressed in SLC38A5-depleted cells and Figure 5A shows that the viability decreased in SLC38A5-depleted cells can be partially rescued by the ferroptosis inhibitor ferrostatin-1. However, I believe that this is not sufficient to state that SLC38A5 directly regulates ferroptosis in these cell lines. Therefore, I suggest that the authors conduct an experiment to observe ferroptosis using a ferroptosis inducer of their choice (such as erastin, RSL3, cysteine-free media etc.) and confirm that the death is blocked by ferroptosis inhibitors of their choice (such as ferrostatin-1, DFO etc.). As the data provided in the current manuscript support this mechanism, I believe the authors may gain positive results that will help support the main finding of this manuscript.

2.     Related to the first comment, the figure title of Figure 3 “Inhibition of SLC38A5 induces ferroptosis in mRNA sequencing” should be modified to “Inhibition of SLC38A5 induces changes in ferroptosis-regulating genes”.

3.     The second major comment is that although expression levels of SLC38A5 correlate well with gemcitabine-resistance as shown in Figure 1 and inhibition of SLC38A5 decreases viability of gemcitabine-resistant cells as shown in Figure 2C and Figure 5A, it is unclear whether SLC38A5 is selectively important for gemcitabine-resistant cells (as gemcitabine-sensitive cells were also shown to decrease viability with SLC38A5 knockdown in Figure 2C). Therefore, I would like to ask the authors if there are any markers that are elevated in gemcitabine-resistant cells in a similar manner as SLC38A5 but the knockdown/inhibition does not decrease viability as SLC38A5 does (perhaps RRM1?). If so, I believe it would be helpful if the authors could provide data for that other target along with SLC38A5 in Figure 2C so that it would support SLC38A5 as a better target.

4.     In Figure 7, the graphical abstract indicates that inhibition of SLC38A5 leads to a loss of GPX4, which is not shown through any data in the manuscript. Therefore, I suggest that the arrows indicating changes in GPX4 should be removed.

Minor comments

1.     Line 400, SLc38A5 should be replaced to SLC38A5.

Author Response

COVER LETTER

Joon Seong Park, MD, PhD

Professor

Department of Surgery, Gangnam Severance Hospital

Yonsei University College of Medicine

The Managing Editor

Cells

Dear Editor,

             Thank you for your letter dated September 18th, 2023, regarding the decision on our manuscript “SLC38A5 modulates ferroptosis to overcome gemcitabine-resistance in pancreatic cancer”, by Kim et al. We would like to thank the reviewers for their rigorous evaluations and valid concerns for our manuscript.

We have carefully discussed the reviewers’ comments and have edited the manuscript accordingly. We found the reviewers’ comments extremely helpful and critical in presenting our findings. We sincerely hope our updated work meets the prestigious standard of the journal.

We submit the Revised Manuscript and this letter, in which we have supplied a list of changes we made to the text in response to each of the individual points supplied by the reviewer. We hope you will see that our original findings are further supplemented by our responses.

Thank you for your consideration, and please do not hesitate to contact me if you have any queries.

Sincerely yours,

Joon Seong Park, MD, PhD

POINT-BY-POINT RESPONSE TO REVIEWERS’ COMMENTS:

Reviewer #1 Comments:

1. The data in Figure 3 collectively shows that genes related to ferroptosis regulation are differentially expressed in SLC38A5-depleted cells and Figure 5A shows that the viability decreased in SLC38A5-depleted cells can be partially rescued by the ferroptosis inhibitor ferrostatin-1. However, I believe that this is not sufficient to state that SLC38A5 directly regulates ferroptosis in these cell lines. Therefore, I suggest that the authors conduct an experiment to observe ferroptosis using a ferroptosis inducer of their choice (such as erastin, RSL3, cysteine-free media etc.) and confirm that the death is blocked by ferroptosis inhibitors of their choice (such as ferrostatin-1, DFO etc.). As the data provided in the current manuscript support this mechanism, I believe the authors may gain positive results that will help support the main finding of this manuscript.

A. Thank you for your insightful comment. We have conducted additional experiments using the ferroptosis inducer erastin. The results have been added to Figure 5A in the Revised Manuscript.

2. Related to the first comment, the figure title of Figure 3 “Inhibition of SLC38A5 induces ferroptosis in mRNA sequencing” should be modified to “Inhibition of SLC38A5 induces changes in ferroptosis-regulating genes”.

A. Thank you for your comment. We have changed the figure title in the Revised Manuscript.

3. The second major comment is that although expression levels of SLC38A5 correlate well with gemcitabine-resistance as shown in Figure 1 and inhibition of SLC38A5 decreases viability of gemcitabine-resistant cells as shown in Figure 2C and Figure 5A, it is unclear whether SLC38A5 is selectively important for gemcitabine-resistant cells (as gemcitabine-sensitive cells were also shown to decrease viability with SLC38A5 knockdown in Figure 2C). Therefore, I would like to ask the authors if there are any markers that are elevated in gemcitabine-resistant cells in a similar manner as SLC38A5 but the knockdown/inhibition does not decrease viability as SLC38A5 does (perhaps RRM1?). If so, I believe it would be helpful if the authors could provide data for that other target along with SLC38A5 in Figure 2C so that it would support SLC38A5 as a better target.

A. Thank you for your comment. We agree with your concern and we have added the PCR of the RRM1 expression to Figure 2C in the Revised Manuscript.

4. In Figure 7, the graphical abstract indicates that inhibition of SLC38A5 leads to a loss of GPX4, which is not shown through any data in the manuscript. Therefore, I suggest that the arrows indicating changes in GPX4 should be removed.

A. Thank you for your comment. We would like to point out that the changes in the expression of GPX4 are highlighted in Figure 4E and 4F. We hope the results we present are satisfactory.

Minor comments

Line 400, SLc38A5 should be replaced to SLC38A5.

A. We have replaced the error in the Revised Manuscript.

Reviewer additional comments:

Grammarly reveals 14% plagiarism. Grammar error 230. The raw data in the supplementary file show WB on a blue blot. The authors need to show the original WB intensity without a blue background
in the manuscript. It is good that they provided the original blots in a supplememtary file.

A. Thank you for your comment. We have reviewed and amended the errors in the Revised Manuscript. We sincerely apologize for the errors in the original script.

We already received a journal article editing for our Manuscript.

Most of the 14% of plagiarism is from papers published by our lab. I am also a co-author of this paper and performed various experiments together. We have added a paper published by our lab as a reference.

Ivermectin and gemcitabine combination treatment induces apoptosis of pancreatic cancer cells via mitochondrial dysfunction, Frontiers in Pharmacology (https://pubmed.ncbi.nlm.nih.gov/36091811/)

The role of LOXL2 induced by glucose metabolism-activated NF-kB n maintaining drug resistance through EMT and cancer stemness in gemcitabine-resistant PDAC, Journal of molecular medicine
(https://pubmed.ncbi.nlm.nih.gov/37737908/)

In addition, the original WB images with the blue background were taken using X-ray films to detect chemiluminescent 2nd antibodies; therefore, the blot images are the original images. The WB images with white backgrounds, on the other hand, were detected digitally using ImageQuant LAS 4000 detector.

We would like to express our sincere appreciation for your thoughtful comments and advice.

Reviewer 2 Report

Summary:

Herein, the authors focus on a glutamine transporter, SLC38A5, and show that it is more overexpressed in gemcitabine-resistant patients than in gemcitabine-sensitive patients. The authors use two human cell lines to show that the knockdown of SLC38A5 decreased the proliferation and migration of gemcitabine-resistant PDAC cells. They show that inhibition of SLC38A5 triggered the ferroptosis signaling pathway, induced mitochondrial dysfunction and reduced glutamine uptake, and glutathione (GSH) levels, and downregulated the expressions of GSH-related genes NRF2 and GPX4. In the same two cells. they further showed that blockade of glutamine uptake negatively modulated the mTOR-SREBP1-SCD1 signaling pathway. Lastly, they observed that the knockdown of SLC38A5 restored gemcitabine sensitivity by hindering tumor growth and metastasis in the orthotopic mouse model.

Major comments:

-          The model system used for the study is only two human cell lines in culture. The authors use nude mice for implanting tumors where there is no immune component. Why were KPC mice or KPC cells not used in this study? This would add more depth and novelty to the findings.

-          In fig1A, why do the mRNA levels of SLC38A5 in PANC1 cells show a reduced band but the protein level is much higher? Also, the protein level in the same cells is more than AsPC1 which does not correlate to the mRNA levels in the above panel. Why?

-          On page 1, line 32, please use the updated or current survival rate/reference.

-          In line 58, the authors state that – “Glutathione is an antioxidant that regulates reactive oxygen species (ROS) levels in cancer cells [15].” The authors must cite an actual research article that made this discovery and not just a review article to give the due credit.

-          On page 5, line 228, the authors suddenly mentioned RRM1. What is RRM1? Why is there no information on this in the introduction section? Why are the authors showing this without any context?

-          BxPC3 (mRNA and protein) had more expression of SLC38A5 than PANC1 (fig1, where the mRNA for SLC38A5 was almost undetectable). Do the authors observe similar results as those in Figure 2 for BxPC3 as well? Why was PANC1 selected and not BxPC3, given the expression results in fig1?

-          In fig2, how did the authors calculate the IC50? There is no section on cell viability in the material and methods section. Please provide details.

-          For Fig, the authors only looked at the OCR and ECAR for two groups. Did they also analyze the results for the Resistant+si group (res+siSLC38A5)?

Minor comments:

-          Please follow journal guidelines for the figure font size (for example for figures 2, 3, and others)

More direct speech and tighter language can be used. But the manuscript is well written in general.

Author Response

COVER LETTER

Joon Seong Park, MD, PhD

Professor

Department of Surgery, Gangnam Severance Hospital

Yonsei University College of Medicine

The Managing Editor

Cells

Dear Editor,

             Thank you for your letter dated September 18th, 2023, regarding the decision on our manuscript “SLC38A5 modulates ferroptosis to overcome gemcitabine-resistance in pancreatic cancer”, by Kim et al. We would like to thank the reviewers for their rigorous evaluations and valid concerns for our manuscript.

We have carefully discussed the reviewers’ comments and have edited the manuscript accordingly. We found the reviewers’ comments extremely helpful and critical in presenting our findings. We sincerely hope our updated work meets the prestigious standard of the journal.

We submit the Revised Manuscript and this letter, in which we have supplied a list of changes we made to the text in response to each of the individual points supplied by the reviewer. We hope you will see that our original findings are further supplemented by our responses.

Thank you for your consideration, and please do not hesitate to contact me if you have any queries.

Sincerely yours,

Joon Seong Park, MD, PhD

POINT-BY-POINT RESPONSE TO REVIEWERS’ COMMENTS:

Reviewer #2 Comments:

  1. The model system used for the study is only two human cell lines in culture. The authors use nude mice for implanting tumors where there is no immune component. Why were KPC mice or KPC cells not used in this study? This would add more depth and novelty to the findings.
    A. Thank you for your comment. We understand your concerns. However, we would like to present previous reports with similar objectives that did not use KPC mice or cells. We have attached the references:

Hsa-miR-3178/RhoB/PI3K/Akt, a novel signaling pathway regulates ABC transporters to reverse gemcitabine resistance in pancreatic cancer, Molecular Cancer (https://pubmed.ncbi.nlm.nih.gov/35538494/)

SRSF3-mediated regulation of N6-methyladenosine modification-related lncRNA ANRIL splicing promotes resistance of pancreatic cancer to gemcitabine, Cell Reports (https://www.sciencedirect.com/science/article/pii/S2211124722005848?via%3Dihub#sec4)

2. In fig1A, why do the mRNA levels of SLC38A5 in PANC1 cells show a reduced band but the protein level is much higher? Also, the protein level in the same cells is more than AsPC1 which does not correlate to the mRNA levels in the above panel. Why?

A. Thank you for the comment. In the image J graph in Figure 1A, the expression of AsPc-1 and PANC-1 is similar. We wanted to show in Figure 1A that SLC38A5 is expressed across most pancreatic cancer cells.

In this study, we emphasize that SLC38A5 is expressed more strongly in resistant cells than in sensitive cells.

3. On page 1, line 32, please use the updated or current survival rate/reference.

A. Thank you for your comment. We have used an updated survival rate in the Revised Manuscript.

4. In line 58, the authors state that – “Glutathione is an antioxidant that regulates reactive oxygen species (ROS) levels in cancer cells [15].” The authors must cite an actual research article that made this discovery and not just a review article to give the due credit.

A. Thank you for your comment. We have changed the reference in the Revised Manuscript.

5. On page 5, line 228, the authors suddenly mentioned RRM1. What is RRM1? Why is there no information on this in the introduction section? Why are the authors showing this without any context?

A. Thank you for your comment. We have added an explanation and reference that state RRM1 as a widely used marker of gemcitabine resistance in the Revised Manuscript. We apologize for not making this point clearer.

6. BxPC3 (mRNA and protein) had more expression of SLC38A5 than PANC1 (fig1, where the mRNA for SLC38A5 was almost undetectable). Do the authors observe similar results as those in Figure 2 for BxPC3 as well? Why was PANC1 selected and not BxPC3, given the expression results in fig1?

A. Thank you for your comment. We selected PANC-1 as our main cell line because, although BxPc-3 showed expression of SLC38A5, the expression of SLC38A5 did not increase in the gemcitabine-resistant BxPc-3 cells.

7. In fig2, how did the authors calculate the IC50? There is no section on cell viability in the material and methods section. Please provide details.

A. Thank you for your comment. We have added the method by which the IC50 was calculated in the Materials and Methods (2.5. WST Assay) in the Revised Manuscript.

8. For Fig, the authors only looked at the OCR and ECAR for two groups. Did they also analyze the results for the Resistant+si group (res+siSLC38A5)?

A. Thank you for your comment. Both OCR and ECAR were measured in gemcitabine-resistant PANC-1 (siControl and siSLC38A5). To avoid any confusion, we have updated the figure legend for Figure 4B in the Revised Manuscript, clearly stating the groups used for the experiment.

Minor comments:

Please follow journal guidelines for the figure font size (for example for figures 2, 3, and others)

A. Thank you for your comment. We have gone back and made sure the journal guidelines were followed in the Revised Manuscript.

We would like to express our sincere appreciation for your thoughtful comments and advice.

Reviewer 3 Report

1. The manuscript reveals over 300 grammar errors with 20% plagiarism which require attention before any consideration of publication.

2. The WB figures in the main body of the manuscript reveal tempering of the images which is not acceptable. The Supplement FIG1A show the raw WB images which should be used.

3. The SLC38A5 protein WB blot in the supplement reveals different strong bands indicating isomers of the protein. Please explain the protein band at 58kDa.

4. In Fig1A, the plot of protein expression derived from the WB shows a decrease in expression but the blot image does not indicate it. Please explain the formula to measure protein expression from the blot bands.

5, Pancreatic tissues were obtained from patients diagnosed with pancreatic cancer at Gangnam Severance Hospital. These tissues require further description with regard to stage of the tumor, etc. The clinical aspects of the patients must be described. (E) Dot plot of SLC38A5 and RRM1 gene expression level in 243 purified mRNA from gemcitabine sensitive and resistance pancreatic cancer patient tissue. Are these the number of patients?

6. The stats asterisks must be consistent. For the in vitro experiments, please indicate how many independent experiments were performed.

7. Microscopic images of the migrated cells were captured - what is the formula to measure wound width? Independent times?

8. The tumors were harvested, weighed, and fixed in 4% paraformaldehyde. Metastases were photographed. Fig 6F only show macro metastases revealing only nodules. The authors must do fixed tissue and embedded in paraffin, sectioned at 5 m, and sections transferred onto glass slides, deparaffinized through xylene and graded alcohols into the water, and stained with hematoxylin and eosin (H&E). Slides  dehydrated and mounted, and analyzed for micro-metastases throughout the tissue.

9. Oxytosis/ferroptosis is a type of programmed cell death dependent on iron and characterized by the accumulation of lipid peroxides. Our results indicate that SLC38A5 deletion results in the accumulation of lipid ROS in gemcitabine-resistant PDAC cells. Figure 5. Inhibition of SLc38A5 induces ferroptosis in gemcitabine-resistance pancreatic cancer cells.(H) FACS analysis of lipid ROS in indicated PANC-GR groups. Bar graph is analyzed by Image J software. Ferrostatin-1 (Fer-1) is treated with 20 uM. What is being treated? Please be precise.

The manuscript reveals over 300 grammar errors with 20% plagiarism which require attention before any consideration of publication.

Author Response

COVER LETTER

Joon Seong Park, MD, PhD

Professor

Department of Surgery, Gangnam Severance Hospital

Yonsei University College of Medicine

The Managing Editor

Cells

Dear Editor,

Thank you for your letter dated September 18th, 2023, regarding the decision on our manuscript “SLC38A5 modulates ferroptosis to overcome gemcitabine-resistance in pancreatic cancer”, by Kim et al. We would like to thank the reviewers for their rigorous evaluations and valid concerns for our manuscript.

We have carefully discussed the reviewers’ comments and have edited the manuscript accordingly. We found the reviewers’ comments extremely helpful and critical in presenting our findings. We sincerely hope our updated work meets the prestigious standard of the journal.

We submit the Revised Manuscript and this letter, in which we have supplied a list of changes we made to the text in response to each of the individual points supplied by the reviewer. We hope you will see that our original findings are further supplemented by our responses.

Thank you for your consideration, and please do not hesitate to contact me if you have any queries.

Sincerely yours,

Joon Seong Park, MD, PhD

POINT-BY-POINT RESPONSE TO REVIEWERS’ COMMENTS:

Reviewer #3 Comments:

  1. The manuscript reveals over 300 grammar errors with 20% plagiarism which require attention before any consideration of publication.
    A. Thank you for your comment. We have reviewed and amended the errors in the Revised Manuscript. We sincerely apologize for the errors in the original script.
    We already received a journal article editing for our Manuscript.
    Most of the 20% of plagiarism is from papers published by our lab. I am also a co-author of this paper and performed various experiments together. We have added a paper published by our lab as a reference.

Ivermectin and gemcitabine combination treatment induces apoptosis of pancreatic cancer cells via mitochondrial dysfunction, Frontiers in Pharmacology (https://pubmed.ncbi.nlm.nih.gov/36091811/)

The role of LOXL2 induced by glucose metabolism-activated NF-kB n maintaining drug resistance through EMT and cancer stemness in gemcitabine-resistant PDAC, Journal of molecular medicine
(https://pubmed.ncbi.nlm.nih.gov/37737908/)

  1. The WB figures in the main body of the manuscript reveal tempering of the images which is not acceptable. The Supplement FIG1A show the raw WB images which should be used.
    A. Thank you for your comment. We apologize for using images that seem tempered. To clarify that the WB images used in the manuscript were not tempered, we have added the original WB images in Supplementary Figure 1B in the Revised Manuscript.

  1. The SLC38A5 protein WB blot in the supplement reveals different strong bands indicating isomers of the protein. Please explain the protein band at 58kDa.
    A. Thank you for your comment. We went back and reviewed the WB images, but we were unable to identify any additional bands. We hope the raw WB images displayed in Supplementary Figure 1B verify that the band at 58 kDa is the main band for SLC38A5.
    Additionally, to avoid further confusion, we added “58 kDa” marks on each of the WB images in Supplementary Figure 1B in the Revised Manuscript.

  1. In Fig1A, the plot of protein expression derived from the WB shows a decrease in expression but the blot image does not indicate it. Please explain the formula to measure protein expression from the blot bands.
    A. Thank you for your comment. To further validate the calculations for the plot in Figure 1A, we attached a snapshot of the calculations used to measure the protein expression.

  1. Pancreatic tissues were obtained from patients diagnosed with pancreatic cancer at Gangnam Severance Hospital. These tissues require further description with regard to stage of the tumor, etc. The clinical aspects of the patients must be described. (E) Dot plot of SLC38A5 and RRM1 gene expression level in purified mRNA from gemcitabine sensitive and resistance pancreatic cancer patient tissue. Are these the number of patients?
    A.Thank you for your comment. We agree with your statement that further patient tissue description is necessary. Therefore, we submit a table describing the clinical aspects of the patient.
    For Figure 1E, the dots do indicate the number of patients. The number of patients in each group is also indicated below the x-axis of the graph.

  1. The stats asterisks must be consistent. For the in vitro experiments, please indicate how many independent experiments were performed.
    A.Thank you for your comment. We have added the phrase “All experiments were performed at least three times” at the end of each figure description in the Revised Manuscript.

  1. Microscopic images of the migrated cells were captured - what is the formula to measure wound width? Independent times?
    A.Thank you for your comment. For the experiment, images were captured at 16 and 24 hours after the cells were plated and the widths were measured using the ImageJ software. The MRI Wound Healing Tool macro was used to measure the sizes of the gap for each image.

  1. The tumors were harvested, weighed, and fixed in 4% paraformaldehyde. Metastases were photographed. Fig 6F only show macro metastases revealing only nodules. The authors must do fixed tissue and embedded in paraffin, sectioned at 5 m, and sections transferred onto glass slides, deparaffinized through xylene and graded alcohols into the water, and stained with hematoxylin and eosin (H&E). Slides dehydrated and mounted, and analyzed for micro-metastases throughout the tissue.
    A.Thank you for your comment. We recognize that our findings lack data that support metastases of the tumor. Therefore, we provide Western and RT-qPCR results that support our findings. In Figure 6C and 6D in the Revised Manuscript, we show that the expression levels E-cadherin and N-cadherin decreased in the knockdown group, but were increased in the ferrostatin-1 group.
    However, unfortunately, as our lab does not have the proper equipment to carry out the H & E imaging within the allotted time for the revision, we were unable to carry out the experiment. We apologize for our inability to conduct the experiment, but we hope that the outcomes we provide will meet your expectations.

  1. Oxytosis/ferroptosis is a type of programmed cell death dependent on iron and characterized by the accumulation of lipid peroxides. Our results indicate that SLC38A5 deletion results in the accumulation of lipid ROS in gemcitabine-resistant PDAC cells. Figure 5. Inhibition of SLc38A5 induces ferroptosis in gemcitabine-resistance pancreatic cancer cells.(H) FACS analysis of lipid ROS in indicated PANC-GR groups. Bar graph is analyzed by Image J software. Ferrostatin-1 (Fer-1) is treated with 20 uM. What is being treated? Please be precise.
    A.Thank you for your comment. When treated, ferrostatin-1 inhibits ferroptosis by suppressing lipid ROS. In this experiment, Ferrostatin-1 (Fer-1) was treated as described: 5 mg of Fer-1 was dissolved in 1.9 mL of DMSO to create a concentration of 10 mM. The solution then was diluted to the respective concentrations before treatment.
    Also, to avoid any further confusion, we have edited the Figure 5H description to “20 µM of Fer-1 was treated to the control and knockdown groups” in the Revised Manuscript.

We would like to express our sincere appreciation for your thoughtful comments and advice.

Reviewer 4 Report

Pancreatic ductal adenocarcinoma (PDAC) ranks as one of the deadliest cancers known. The current standard of care for advanced PDAC involves a combination therapy utilizing nab-paclitaxel and gemcitabine. Nevertheless, a significant portion of patients eventually develop resistance to gemcitabine. In this manuscript, the authors explored the crucial role of SLC38A5 in gemcitabine-resistant pancreatic cancer. This study revealed elevated expression levels of SLC38A5 in both PDAC and gemcitabine-resistant cell lines, suggesting its involvement in the resistance mechanism. Notably, SLC38A5 appears to function through the ferroptosis-related signaling pathway. This discovery positions SLC38A5 as a promising target for PDAC therapy, particularly for patients resistant to gemcitabine treatment.

Key Comments on this Manuscript:

Other research groups have also investigated the role of SLC38A5 in PDAC. We should discuss the differences and unique contributions of our study in comparison to theirs. https://doi.org/10.1158/1940-6215.PrecPrev22-P005

In Figure 1B and D, you explored the expression and prognosis of SLC38A5 in PDAC in contrast to normal tissue. However, you should also include data comparing gemcitabine-resistant patients to those without resistance/ normal controls for a more comprehensive analysis.

Please ensure that the scale is appropriately labeled in Figure 1C.

In Figure 2A, it would be beneficial to include labels indicating the concentration of IC50.

Don't forget to label the p-value and FDR in Figure 3C for clarity.

To further support the  findings, it is advisable to evaluate certain ferroptosis-related markers in vivo through techniques such as western blotting or immunohistochemistry (IHC), as depicted in Figure 6.

None

Author Response

COVER LETTER

Joon Seong Park, MD, PhD

Professor

Department of Surgery, Gangnam Severance Hospital

Yonsei University College of Medicine

The Managing Editor

Cells

Dear Editor,

             Thank you for your letter dated September 18th, 2023, regarding the decision on our manuscript “SLC38A5 modulates ferroptosis to overcome gemcitabine-resistance in pancreatic cancer”, by Kim et al. We would like to thank the reviewers for their rigorous evaluations and valid concerns for our manuscript.

We have carefully discussed the reviewers’ comments and have edited the manuscript accordingly. We found the reviewers’ comments extremely helpful and critical in presenting our findings. We sincerely hope our updated work meets the prestigious standard of the journal.

We submit the Revised Manuscript and this letter, in which we have supplied a list of changes we made to the text in response to each of the individual points supplied by the reviewer. We hope you will see that our original findings are further supplemented by our responses.

Thank you for your consideration, and please do not hesitate to contact me if you have any queries.

Sincerely yours,

Joon Seong Park, MD, PhD

POINT-BY-POINT RESPONSE TO REVIEWERS’ COMMENTS:

Reviewer #4 Comments:

  1. Other research groups have also investigated the role of SLC38A5 in PDAC. We should discuss the differences and unique contributions of our study in comparison to theirs. https://doi.org/10.1158/1940-6215.PrecPrev22-P005
    A.Thank you for your comment. We have reviewed the previous study, and we recognize the important contributions made by the study.
    However, while the previous study investigated the role of SLC38A5 in PDAC, our study uniquely focused on the role of SLC38A5 in gemcitabine-resistant PDAC. We report that SLC38A5 is highly expressed in gemcitabine-resistant patient tissue and PDAC cell lines. Our findings distinctively show that knocking down SLC38A5 in gemcitabine-resistant cells reduced the resistant properties of the resistant cells.
  1. In Figure 1B and D, you explored the expression and prognosis of SLC38A5 in PDAC in contrast to normal tissue. However, you should also include data comparing gemcitabine-resistant patients to those without resistance/ normal controls for a more comprehensive analysis.
    A.Thank you for your comment. We agree that adding the patient WB data comparing gemcitabine-resistant to those without resistance to Figure 1B can more accurately depict our conclusions. However, Figure 1F shows the discrepancy in the expression of SLC38A5 between gemcitabine-resistant and non-resistant patient tissues. Therefore, we believe that Figure 1 as a whole depicts a comprehensive analysis of the expression of SLC38A5 in gemcitabine-resistant PDAC.
    For the survival analysis in Figure 1D, we utilized open data; therefore, we are unable to include the survival data comparing gemcitabine-resistant patients to those without resistance. We apologize for being unable to procure this analysis.
  1. Please ensure that the scale is appropriately labeled in Figure 1C.
    A.Thank you for your comment. We have added the scale bars in the figure of the Revised Manuscript.
  2. In Figure 2A, it would be beneficial to include labels indicating the concentration of IC50.
    A.Thank you for your comment. We have added the unit of measurement for the IC50 values (µM) in Figure 2A of the Revised Manuscript. In addition, we have changed the x-axis to indicate the concentrations of gemcitabine. We apologize if the data caused any confusion.

  1. Don't forget to label the p-value and FDR in Figure 3C for clarity.
    A.Thank you for your comment. The p-values have been added in Figure 3C of the Revised Manuscript.

  1. To further support the findings, it is advisable to evaluate certain ferroptosis-related markers in vivo through techniques such as western blotting or immunohistochemistry (IHC), as depicted in Figure 6.
    A.Thank you for your insightful comment. We have added the western blot analysis of ferroptosis-related markers from in vivo tumors in Figure 6C and 6D of the Revised Manuscript.

We would like to express our sincere appreciation for your thoughtful comments and advice.

Round 2

Reviewer 1 Report

Thank you for the revisions throughout the manuscript.

Author Response

Thank you for your review.

Reviewer 2 Report

  1. The model system used for the study is only two human cell lines in culture. The authors use nude mice for implanting tumors where there is no immune component. Why were KPC mice or KPC cells not used in this study? This would add more depth and novelty to the findings.

Response: Thank you for your comment. We understand your concerns. However, we would like to present previous reports with similar objectives that did not use KPC mice or cells. We have attached the references:

Hsa-miR-3178/RhoB/PI3K/Akt, a novel signaling pathway regulates ABC transporters to reverse gemcitabine resistance in pancreatic cancer, Molecular Cancer (https://pubmed.ncbi.nlm.nih.gov/35538494/)

SRSF3-mediated regulation of N6-methyladenosine modification-related lncRNA ANRIL splicing promotes resistance of pancreatic cancer to gemcitabine, Cell Reports (https://www.sciencedirect.com/science/article/pii/S2211124722005848?via%3Dihub#sec4)

Query: This is not a valid response and is not justified.

2. In fig1A, why do the mRNA levels of SLC38A5 in PANC1 cells show a reduced band but the protein level is much higher? Also, the protein level in the same cells is more than AsPC1 which does not correlate to the mRNA levels in the above panel. Why?

Response. Thank you for the comment. In the image J graph in Figure 1A, the expression of AsPc-1 and PANC-1 is similar. We wanted to show in Figure 1A that SLC38A5 is expressed across most pancreatic cancer cells.

In this study, we emphasize that SLC38A5 is expressed more strongly in resistant cells than in sensitive cells.

Query: The authors do not explain why the mRNA levels of SLC38A5 in PANC1 cells show a reduced band but the protein level is much higher. Also, are the authors implying that the protein band (western blot) for SLC38A5 in PANC1 cells is equal to the AsPC1?

4. In line 58, the authors state that – “Glutathione is an antioxidant that regulates reactive oxygen species (ROS) levels in cancer cells [15].” The authors must cite an actual research article that made this discovery and not just a review article to give the due credit.

Response. Thank you for your comment. We have changed the reference in the Revised Manuscript.

Query: The cited reference does not reflect the actual research article which demonstrated that “Glutathione is an antioxidant that regulates reactive oxygen species (ROS) levels in cancer cells.” Please provide the actual article.

6. BxPC3 (mRNA and protein) had more expression of SLC38A5 than PANC1 (fig1, where the mRNA for SLC38A5 was almost undetectable). Do the authors observe similar results as those in Figure 2 for BxPC3 as well? Why was PANC1 selected and not BxPC3, given the expression results in fig1?

Response. Thank you for your comment. We selected PANC-1 as our main cell line because, although BxPc-3 showed expression of SLC38A5, the expression of SLC38A5 did not increase in the gemcitabine-resistant BxPc-3 cells.

Query: The authors must provide this data on BxPC3 and the explanation above in the manuscript.

Fine

Author Response

COVER LETTER

Joon Seong Park, MD, PhD

Professor

Department of Surgery, Gangnam Severance Hospital

Yonsei University College of Medicine

The Managing Editor

Cells

Dear Editor,

             Thank you for your letter dated October 13th, 2023, regarding the decision on our manuscript “SLC38A5 modulates ferroptosis to overcome gemcitabine-resistance in pancreatic cancer”, by Kim et al. We would like to thank the reviewers for their rigorous evaluations and valid concerns for our manuscript.

We have carefully discussed the reviewers’ comments and have edited the manuscript accordingly. We found the reviewers’ comments extremely helpful and critical in presenting our findings. We sincerely hope our updated work meets the prestigious standard of the journal.

We submit the Revised Manuscript and this letter, in which we have supplied a list of changes we made to the text in response to each of the individual points supplied by the reviewer. We hope you will see that our original findings are further supplemented by our responses.

Thank you for your consideration, and please do not hesitate to contact me if you have any queries.

Sincerely yours,

Joon Seong Park, MD, PhD

POINT-BY-POINT RESPONSE TO REVIEWERS’ COMMENTS:

Reviewer #2 Comments:

  1. The model system used for the study is only two human cell lines in culture. The authors use nude mice for implanting tumors where there is no immune component. Why were KPC mice or KPC cells not used in this study? This would add more depth and novelty to the findings.

1st revision A. Thank you for your comment. We understand your concerns. However, we would like to present previous reports with similar objectives that did not use KPC mice or cells. We have attached the references:

Hsa-miR-3178/RhoB/PI3K/Akt, a novel signaling pathway regulates ABC transporters to reverse gemcitabine resistance in pancreatic cancer, Molecular Cancer (https://pubmed.ncbi.nlm.nih.gov/35538494/)

SRSF3-mediated regulation of N6-methyladenosine modification-related lncRNA ANRIL splicing promotes resistance of pancreatic cancer to gemcitabine, Cell Reports (https://www.sciencedirect.com/science/article/pii/S2211124722005848?via%3Dihub#sec4)

Query: This is not a valid response and is not justified.

2nd revision A. Thank you for your comment. As you have pointed out, the KPC mouse model would have added more depth to our research.  However, as KPC mice were costly and time consuming, we were unable to allocate the proper funds and time for the experiments.

Additionally, we believed that nude mouse model was a better fit for this study, as we do not discuss the impact of the immune component in gemcitabine-resistant PDAC cells. We have attached a reference that states that nude mice models are widely used as orthotopic models of human cancer cells. Thank you again for your thoughtful comments. 

Modeling pancreatic cancer in mice for experimental therapeutics, Biochim Biophsy Acta Rev Cancer
(https://pubmed.ncbi.nlm.nih.gov/33945847/)

  1. In fig1A, why do the mRNA levels of SLC38A5 in PANC1 cells show a reduced band but the protein level is much higher? Also, the protein level in the same cells is more than AsPC1 which does not correlate to the mRNA levels in the above panel. Why?

1st revision A. Thank you for the comment. In the image J graph in Figure 1A, the expression of AsPc-1 and PANC-1 is similar. We wanted to show in Figure 1A that SLC38A5 is expressed across most pancreatic cancer cells.

In this study, we emphasize that SLC38A5 is expressed more strongly in resistant cells than in sensitive cells.

Query: The authors do not explain why the mRNA levels of SLC38A5 in PANC1 cells show a reduced band but the protein level is much higher. Also, are the authors implying that the protein band (western blot) for SLC38A5 in PANC1 cells is equal to the AsPC1?

2nd revision A. Thank you for your comment. We performed the PCR experiment again and found that the results matched the western blot results. We believe there was a technical error in the previous result, and we have updated the result in the Revised Manuscript. We apologize for the confusion.

  1. In line 58, the authors state that – “Glutathione is an antioxidant that regulates reactive oxygen species (ROS) levels in cancer cells [15].” The authors must cite an actual research article that made this discovery and not just a review article to give the due credit.

1st revision A. Thank you for your comment. We have changed the reference in the Revised Manuscript.

Query: The cited reference does not reflect the actual research article which demonstrated that “Glutathione is an antioxidant that regulates reactive oxygen species (ROS) levels in cancer cells.” Please provide the actual article.

2nd revision A. Thank you for your comment. We have changed the reference in the Revised Manuscript.

Regulation of ferroptotic cancer cell death by GPX4, Cell
( https://www.cell.com/fulltext/S0092-8674(13)01544-4 )

  1. BxPC3 (mRNA and protein) had more expression of SLC38A5 than PANC1 (fig1, where the mRNA for SLC38A5 was almost undetectable). Do the authors observe similar results as those in Figure 2 for BxPC3 as well? Why was PANC1 selected and not BxPC3, given the expression results in fig1?

1st revision A. Thank you for your comment. We selected PANC-1 as our main cell line because, although BxPc-3 showed expression of SLC38A5, the expression of SLC38A5 did not increase in the gemcitabine-resistant BxPc-3 cells.

Query: The authors must provide this data on BxPC3 and the explanation above in the manuscript.

2nd revision A. Thank you for your comment. Our PCR results shown below indicate that SLC38A5 did not increase significantly in BxPC-3. We also have added an explanation addressing the matter in the Revised Manuscript.

Thank you again for your insightful comments.

Reviewer 3 Report

Improved manscript

minor editing

Author Response

Thank you for your review.